# The Outstanding Chemodiversity of Marine-Derived *Talaromyces*

**DOI:** 10.3390/biom13071021

**Published:** 2023-06-21

**Authors:** Rosario Nicoletti, Rosa Bellavita, Annarita Falanga

**Affiliations:** 1Council for Agricultural Research and Economics, Research Center for Olive, Fruit and Citrus Crops, 81100 Caserta, Italy; 2Department of Agricultural Sciences, University of Naples Federico II, 80055 Portici, Italy; annarita.falanga@unina.it; 3Department of Pharmacy, University of Naples Federico II, 80100 Napoli, Italy; rosa.bellavita@unina.it

**Keywords:** *Talaromyces*, marine-derived fungi, *Penicillium* subgenus *Biverticillium*, bioactive compounds, biotechnological applications, chemotaxonomy

## Abstract

Fungi in the genus *Talaromyces* occur in every environment in both terrestrial and marine contexts, where they have been quite frequently found in association with plants and animals. The relationships of symbiotic fungi with their hosts are often mediated by bioactive secondary metabolites, and *Talaromyces* species represent a prolific source of these compounds. This review highlights the biosynthetic potential of marine-derived *Talaromyces* strains, using accounts from the literature published since 2016. Over 500 secondary metabolites were extracted from axenic cultures of these isolates and about 45% of them were identified as new products, representing a various assortment of chemical classes such as alkaloids, meroterpenoids, isocoumarins, anthraquinones, xanthones, phenalenones, benzofurans, azaphilones, and other polyketides. This impressive chemodiversity and the broad range of biological properties that have been disclosed in preliminary assays qualify these fungi as a valuable source of products to be exploited for manifold biotechnological applications.

## 1. Introduction

The genus *Talaromyces* (Eurotiomycetes, Trichocomaceae) was established about 70 years ago to classify the teleomorphs of some *Penicillium* species [1]. It was primarily considered to include soil fungi after the type species *T. flavus* was mainly reported and exploited as an antagonist of soil-borne plant pathogens [2]. However, as investigations within natural contexts progressed, *Talaromyces* species were found to occur in every environment and be associated not only with terrestrial organisms, such as plants [3] and insects [4], but also to be widespread at sea.

Symbiotic relationships involving fungi are often mediated by their extraordinary capacity for synthesizing bioactive compounds, playing either a promoting or a detrimental role toward the host [5,6]. This is the case for *Talaromyces* species too, based on the high number of reports in the literature [7,8,9,10]. Following a review on the bioactive products of the marine-derived strains of these fungi published at the beginning of 2016 [11], this paper examines the biosynthetic capacities of *Talaromyces* strains recovered from marine sources based on the pertinent literature published since then, in view of providing an updated account on the chemodiversity of these fungi in relation to their possible biotechnological applications.

## 2. Occurrence of *Talaromyces* in the Marine Environment

Deeply rooted in our culture, considering land and sea as separate worlds, the concept of grouping in the two broad categories of terrestrial and marine organisms is basically referable to higher animals and plants, the species of which are generally adapted to either one or the other of these macroecological contexts. However, despite attempts by early marine mycologists, such separation has not proved to be effective in the case of fungi; indeed, a multitude of fungal species originally described from terrestrial sources have been later reported in marine contexts [11,12,13].

Resulting from an increasing number of studies worldwide, the species in the genus *Talaromyces* represent a good example of this adaptability. In fact, an examination of the literature published since 2016 yielded 95 reports of about 30 species (Table 1). Two of them (*T. haitouensis* and *T. zhenhaienis*) were not previously identified in terrestrial contexts, representing a further indication that these fungi are not merely occasional in marine environments. In 30 cases, the isolates were not identified at the species level (reported as *Talaromyces* sp. in Table 1), which could imply an even higher species diversity. Indeed, the issue of species identification for *Talaromyces* is quite fickle, following the recent spread of biomolecular tools in fungal taxonomy and the ensuing nomenclatural revisions. Currently, there are over 170 accepted species in this genus, which are grouped into seven sections [14,15]; four of them (*Helici, Islandici, Talaromyces,* and *Trachyspermi*) include the species listed in Table 1. Besides the most common species, i.e., *T. purpureogenus, T. verruculosus, T. stipitatus, T. pinophilus,* and *T. funiculosus,* the new species *T. haitouensis* and *T. zhenhaienis* also belong to the section *Talaromyces* [16]. The species *T. cellulolyticus* and *T. variabile* have been reported in synonymy with *T. pinophilus* and *T. wortmannii*, respectively [14]; however, in Table 1 we used their old names to avoid possible confusion. Additionally, we provisionally considered the species name *T. cyanescens,* even if it is not included in the updated list of accepted *Talaromyces* species [15]. Several strains examined in this review were identified by the authors with reference to the old *Penicillium* nomenclature [17,18,19,20,21,22,23,24,25,26,27,28,29,30]; however, their identity as *Talaromyces* has been confirmed by morphological descriptions and/or a blast of their DNA sequences in GenBank.

The sources of isolation of these marine-derived *Talaromyces* (Table 1) are diverse, including sediments, water samples, and a variety of plants and animals, within which no specific symbiotic association can be inferred for the time being. As for their geographic origins, it is quite impressive that about 80% of these findings come from Asia and about half from China, which undoubtedly reflects a higher attention paid to the issue of biodiversity by researchers in this area. Reports from ocean trenches and Antarctica further confirm the extraordinary adaptability of these fungi to extreme environmental conditions.

## 3. Structural Aspects

Our overview of the pertinent literature published since 2016 yielded a list of as many as 514 compounds that are reported as secondary metabolites of marine-derived *Talaromyces* strains, resulting from the combination of 230 novel and 284 known products (Table 2 and Table 3). Such an impressive chemodiversity originates from comprehensive genetic bases driving various biosynthetic pathways and assorted biogenic schemes, so that the classification of some structurally complex compounds in a defined chemotype is problematic. Therefore, our attempt to group these products into classes, as indicated in Table 2 and Table 3, is affected by some approximations for a few compounds presenting complex structures.

The remarkable number of new compounds resulting from the biochemical characterization of *Talaromyces* strains show some degree of specificity, possibly reflecting chemotaxonomic relevance. In this respect, the novel products displaying uncommon scaffolds require verification for the possible occurrence of structural analogs in other fungi; however, after many years of study, some compounds have been found exclusively or almost exclusively in *Talaromyces* and can be considered as candidates for the assessment of phylogenetic relations. This is the case of funicone-like compounds, which are characterized by a molecular structure that is built on a γ-pyrone ring linked through a ketone group to an α-resorcylic acid nucleus (Figure 1 (**1**)); besides the true funicones, the other products in this series present modifications on the α-resorcylic acid nucleus, the γ-pyrone ring, or both moieties, and are grouped into the phthalide, furopyrone, and pyridone subclasses [10,112,113]. Among the widely represented oxaphenalenones, duclauxins present diverse polycylic skeletons, generally containing a common dihydrocoumarin benzo[de]isochromen-1(3*H*)-one moiety (Figure 1 (**2**)), while bacillisporins are based on a conjugated 6/6/6/5/6/6/6 ring system (Figure 1 (**3**)), and, in duclauxamides, the ester in one monomer is replaced by an amide group (Figure 1 (**4**)). Multiple polycyclic bridged frames can be found in other products from this class, such as verruculosins (Figure 1 (**5**)), talaromycesones (Figure 1 (**6**)), and macrosporusones (Figure 1 (**7**)) [81,114]. Other typical *Talaromyces* secondary metabolites are mitorubrins (Figure 1 (**8**)) [115], *N*-(4-hydroxy-2-methoxyphenyl)acetamide (Figure 1 (**9**)), and chrodrimanins (Figure 1 (**10**)) [37]. Indeed, after the taxonomic framework is more accurately set following the recent revisions and improvements in the identification procedures, it is to be expected that a thorough examination of the biochemical properties of the accepted taxa may help in considering a number of products as possible chemotaxonomic markers, even for species discrimination within the genus *Talaromyces*.

Other products are representative of widespread classes of organic compounds. Besides being common in plants, isocoumarins have been reported as secondary metabolites in many fungi [116]; nevertheless, twenty new compounds of this type have been described from eight marine-derived *Talaromyces* strains. Azaphilones are another class of typical fungal secondary metabolites [117] that have been particularly investigated as products of *Talaromyces* strains of marine origin, representing one of the most credited sources of these pigments [35,96,110,111]. Likewise, anthraquinones and the related xanthones have also found application as dyes, but their more widespread occurrence in plants has, so far, diminished the appeal of this fungal source [118,119]; however, new products from these classes have been characterized from marine-derived strains of *T. islandicus* [93], *T. stipitatus* [83], and *Talaromyces* sp. [74].

Also widespread among fungi, meroterpenoids are inclusive of very diverse compounds with complex structures of mixed biogenic origin [120]. As such, it is not surprising that the chemosynthetically versatile *Talaromyces* spp. may be able to produce a wide array of these compounds, with a variety of novel structural models. This is the case of talaromyolides A and D (Figure 2 (**11**,**12**)), which present two novel carbon skeletons [58]. Taladrimanin A (Figure 2 (**13**)) represents the first drimane-type meroterpenoid, with a C10 polyketide unit bearing an 8*R* configuration [78]. The above-mentioned chrodrimanins include chlorinated (chrodrimanins K and L) and trichlorinated (chrodrimanin O) versions (Figure 2 (**14**–**16**)), with the latter displaying a unique dichlorine functionality [21,101]. The related amestolkolides A–D (Figure 2 (**17**,**18**)) present a congested pentacyclic skeleton [100], while talaromynoids A, G, H, and I (Figure 2 (**19**–**22**)) possess unprecedented 5/7/6/5/6/6, 6/7/6/6/6/5, 6/7/6/5/6/5/4, and 7/6/5/6/5/4 polycyclic systems, respectively [59]. Other peculiar compounds have been identified among terpenoids, such as talascortene A (Figure 2 (**23**)), a cleistanthane-type diterpenoid possessing a chlorine atom in a peculiar position [64]; moreover, diolhinokiic acid (Figure 2 (**24**)) is the first thujopsene-type sesquiterpenoid containing a 9,10-diol moiety, while roussoellol C (Figure 2 (**25**)) possesses a novel tetracyclic fusicoccane framework with an unexpected hydroxyl at C-4 [109]. Finally, talasteroid (Figure 2 (**26**)) is a new withanolide with a 4-substituted 2,3-dimethyl-2-butenolide ring in its side chain [84].

Structural elucidation has also disclosed some rare or unique molecular scaffolds in other classes. Talaropeptins A and B (Figure 3 (**27**,**28**)) are two new tripeptides that have been identified as products of a non-ribosomal peptide synthase gene cluster, presenting an unusual heterocyclic scaffold and *N*-*trans*-cinnamoyl moiety [108]. The new penixanthones C–D (Figure 3 (**29**,**30**)) also display an unprecedented polycyclic scaffold [90]. Talarodrides A–D (Figure 3 (**31**–**34**)) share a rare caged bicyclo-decadiene with a bridgehead olefin and maleic anhydride core skeleton, while the first case of a naturally occurring 5/7/6 methanocyclononafuran skeleton can be observed in talarodrides E–F (Figure 3 (**35**,**36**)) [72]. The oxidized tricyclic system of talaramide A (Figure 3 (**37**)) has been found for the second time in alkaloids [91]. From a strain of *T. mangshanicus*, talaromanloid A (Figure 3 (**38**)), talaromydene (Figure 3 (**39**)), and ditalaromylectones A–B (Figure 3 (**40**,**41**)) show novel carbon scaffolds; in particular, ditalaromylectone A is a dimeric molecule of 10-hydroxy-8-demethyltalaromydine and dioxo-propanylidene-pyrrolidinyl acrylic acid, while ditalaromylectone B is a cyclized dimer of hydroxydemethyltalaromydines [51]. Talabenzofurans A–B (Figure 3 (**42**,**43**)) possess a peculiar thioester moiety derived from benzofuran and 2-hydroxy-3-mercaptopropionic acid, which is rarely observed in natural products [76]. Novel structural features have also been reported in the typical classes of funicones, with pinophilones A–B (Figure 3 (**44**,**45**)) showing a dihydrofuran moiety for the first time in these compounds [26], and oxaphenalenones. Among the latter, talaromyoxaones A–B (Figure 3 (**46**,**47**)) present a hemiacetal frame and an unprecedented spiro-isobenzofuran-pyranone unit showing biosynthetic enantiodivergence [60]. Finally, the new polyketides, penitalarins A–C (Figure 3 (**48**–**50**)), with a 3,6-dioxabicyclo(3.1.0)hexane ring, are likely a result of synergistic biosynthesis; in fact, they were identified from co-cultures of two strains of *T. aculeatus* and *T. variabile*, while none of them was found when the two strains were cultured independently [24].

Other compounds have proved to be analogs of known products, bringing to their structural revision. For instance, NMR data indicated that talaromyacin A (Figure 3 (**51**)) [95] is identical to sequoiamonascin A, which was originally reported from an endophytic strain of *Aspergillus parasiticus* [121]. Likewise, talacyanol C (Figure 3 (**52**)), from a strain of *T. cyanescens* [38], corresponds to a diastereoisomer of pinophol A, a polyene previously identified as a product of a strain of *T. pinophilus* endophytic in *Salvia miltiorrhiza* [122].

Probably the best example of the chemodiversity in *Talaromyces* is represented by strain G59 of *T. purpureogenus* (generally referred to in the literature as *Penicillium purpurogenum*). In fact, its biosynthetic potential has been explored through the induction of mutants and the activation of silent biosynthetic pathways, by means of neomycin and diethylsulphate, which led to the identification of a long series of compounds. With reference to products identified after 2015, this list includes five cyclic dipeptides, including the novel penicimutide [18]; a novel oxaphenalenone, penicimutalidine, along with the known SF226, bacillisporin C, and corymbiferan lactone A [104]; the novel cyclopentachromone sulfide chromosulfine [102]; the rare carbamate-containing prenylated indole alkaloids penicimutamides A–E [105,106]; the new diketopiperazine derivatives penicimutanolones A–B, penicimutanolone A methyl ether, penicimumide [56], penicimutanin C, and the known penicimutanin A, fructigenines A–B, and rugulosuvine A [107]; the known azaphilones (-)-mitorubrin and (-)-mitorubrinol, isolated along with the new polyketide purpurogenic acid [99]; two new polyketides, purpurofuranone and purpuropyranone, and the known cillifuranone and taiwapyrone [98].

### 3.1. Biogenesis and Structure-Activity Relationships

Some clues on the biogenic origins of secondary metabolites have been gathered by the research activity on marine-derived *Talaromyces*. For instance, 6-hydroxymellein was identified as a possible precursor in the synthesis of meroterpenoids, such as taladrimanin A [78], talaromytin, and the talaromyolides [58]. Other meroterpenoids are presumed to be derived from aromatic polyketide 3,5-dimethylorsellinic acid, such as the talaromynoids [59], amestolkolides, and their related compounds [100], while orsellinic acid is considered to be the biogenic precursor of talabenzofurans and eurothiocins [76], as well as compounds in the funicone series [112,113]. A biosynthetic pathway was proposed for the alkaloid talaramide A, which involves acetyl, malonic acid, and l-leucine as possible precursors [91]. Finally, the joint isolation of benzophenones and xanthones as products of a strain of *T. islandicus* is considered to support the hypothetic biogenesis of xanthones via a benzophenone intermediate [49].

The finding of series of analog compounds differing in certain molecular substitutions has allowed comparative hypotheses concerning bioactivities. Questinol, citreorosein, and fallacinol (Figure 4 (**53**–**55**)) are structurally similar anthraquinones, in which hydroxyl groups have been determined to be essential for their reported anti-obesity activities. In fact, a replacement of the hydroxyls at C-1 (as in questinol) or C-3 (as in fallacinol) by a methoxy group diminishes or completely removes this kind of bioactivity [83]. Moreover, the increasing molecular polarity and hydroxylation of the non-aromatic carbons in structures of anthraquinones was found to strengthen their antibacterial effects, but to weaken their antioxidant activity [93]. The hydroxy group on the benzene ring is also essential for the antioxidant properties of talamins A and D (Figure 4 (**56**,**57**)) [52]. The methylation of the carboxylic group of peniphenone (Figure 4 (**58**)) reduces its immunosuppressive activity; moreover, the immunosuppressive properties of sydowinin A and pinselin are, respectively, higher than those of sydowinin B and hydroxy-methyl-oxo-xanthene-carboxylate (Figure 4 (**59**–**62**)), indicating that the hydroxyl group at C-2 is relevant for this activity [17]. The antibacterial activity of trihydroxy-methoxy-methylbenzophenone (Figure 4 (**63**)) was found to be weakened by methoxylation at C-3 [49]. Conversely, the methylation of 14-OH likely enhances the antibacterial activity of talascortenes (Figure 2 (**23**)) [64]. Likewise, among talarodrides, the higher antibacterial performance of talarodride B (Figure 3 (**32**)) is indicative of the key role played by its methoxy group [72]. Among isocoumarins (Figure 4 (**64**)), aspergillumarin B (Figure 4 (**65**)), with a hydroxy group at C-13, shows no antibacterial activity, unlike other members of this class, such as aspergillumarin A (Figure 4 (**66**)), peniciisocoumarin D, and penicilloxalone B, presenting a keto group in this position; this is indicative of a relevant role of the latter in the bioactivity of these compounds [70]. Again, the presence of two keto carbonyl groups at C-10 and C-13 in amestolkolide B (Figure 4 (**67**)) is thought to enhance its anti-inflammatory effects, in addition to the role of its epoxy group as an active function, which is known to easily react with nucleophiles by ring opening [100].

In another case, the strong α-glucosidase inhibitory effect of eurothiocin D (Figure 4 (**68**)) is presumed to be derived from *α*-d-glucopyranosyl unit substitution, which likely supports its interaction with the enzyme. Moreover, a hydrophilic terminal of the isopentenyl group plays an important role in α-glucosidase inhibition [48]. The presence of a lactone ring and hydroxyl at C-10 is crucial for the antimicrobial activity of the depsidone derivatives talaronins A–E (Figure 4 (**69**–**73**)), which are considered as promising leads against *Helicobacter pylori* [77]. The dimethylcyclobutanol subunit has been proposed as relevant for the antiviral activity of talaromyolide D (Figure 2 (**12**)), making it a valuable target for biosynthetic studies [58]. Furthermore, the dimeric oxyphenalenone scaffold has proved to be essential for the antibacterial and antibiofilm activities of bacillisporins; moreover, the acetoxy group in bacillisporin A has been determined to potentiate bioactivity in comparison with bacillisporin B (Figure 1 (**3**)), bearing a hydroxyl at this position [54]. Finally, comparative assessments concerning mangrovamide A (Figure 4 (**74**)) and its 11,17-*epi*-isomer have indicated a higher antibacterial activity when both C-11 and C-17 are in *R* configuration [45].

### 3.2. Other Biological Sources of the Known Compounds

The data presented in the previous section are indicative of the quite original biosynthetic capacities of *Talaromyces* species/strains, to such an extent that even a good proportion of the known secondary metabolites (Table 3) were first identified from these fungi. Besides the previously mentioned funicones, vermistatins, oxaphenalenones, chrodrimanins, verruculides, mitorubrins, and related azaphilones, this share includes compounds such as deoxyrubralactone, the mangrovamides, miniolutelide C, penicillide and its related products, hydroxypentacecilide, penicifuran, the purpuresters, purpurogenolide E, purpurquinone, the talaromycins, thailandolide B, and wortmin [8,11]. Moreover, the coculnols, which are structurally related to penicillic acid, were originally found in co-cultures of a strain of *Talaromyces* sp. and a strain of *Fusarium solani* [123]. Penicillide appears to be the most common of the above products; in fact, it was identified from five isolates of different species, besides being previously reported from a few more marine-derived strains [11] and being quite frequent among terrestrial *Talaromyces*, too [10,124]. Whether or not this product has implications in the biosynthesis of other secondary metabolites deserves circumstantial studies.

Several products in Table 3 are of a general occurrence among fungi and have been reported to represent biosynthetic intermediates or perform a structural role. This is the case of tyrosol, melleins, benzaldehyde, benzoic, mevalonic and orsellinic acid derivatives, and ergosterols.

Many secondary metabolites were first identified from the phylogenetically related *Penicillium* and *Aspergillus*, which is indicative of a partly common genetic background. In fact, compounds such as alantrypinone, berkedrimane B, the berkeleyacetals, cillifuranone, corymbiferan lactone A, the expansols, the fructigenines, penicilloxalone B, penicillquei A, penioxalicin, pinselin, questin, questinol, rugulosin, rugulosuvine, and the secalonic acids have been previously reported from *Penicillium* species [11,125], while aspergilactone B, the aspergillumarins, asperitaconic acid, the austins, azaspirofuran A, the carnemycins, dihydroaspyrone, diorcinol, eurothiocin A, flavuside B, fonsecinone A, nafuredin, the pseurotins, sequoiamonascin C, similanpyrone B, the sydowinins, terrein, and territrem B are primarily known as *Aspergillus* secondary metabolites [7,126]. In particular, a long series of compounds were first identified from *A. fumigatus*, including fumigaclavine, the fumiquinazolines, fumiquinone B, fumigatin oxide, helvolic acid, trypacidin, the tryptoquivalines, and tryprostatin derivatives, in connection with the thorough investigational activity carried out around this human pathogenic species [127].

Other products are well known or were first identified from other fungi. Some of these are more commonly reported as secondary metabolites of important genera, such as *Fusarium,* known as producer of naphthoquinones [128,129], along with the decalin polyketide fusarielin M [130] and trichothecene solaniol [131], while altenusin and alternaphenol are quite commonly reported among *Alternaria* mycotoxins [132]. On the other hand, many compounds are apparently less renowned since they are reported from fungi of a lower ecological or economic impact. This is the case of chaetominine and rheoemodin from *Chaetomium* spp. [133]; ramulosins from *Pestalotia ramulosa* (currently *Truncatella angustata*) [134]; xylapyrone E from an endophytic *Xylaria* sp. [135]; leptosphaerin G, which is structurally related to secalonic acids, from a strain of *Leptosphaeria* sp. [136]; sclerotinins, characterized as plant growth promoters from *Sclerotinia sclerotiorum* [137]; the alkaloid premalbrancheamide from *Premalbranchea aurantiaca* [138]; sordarin, which is better known as an antifungal product from *Podospora (=Sordaria) araneosa* [139]; taiwapyrone from *Cercospora taiwanensis* [140]; and piniterpenoid D from the fruit bodies of the basidiomycete *Phellinus pini* [141]. Moreover, ethyl everninate was originally identified from the lichen *Evernia prunastri* [142], while nodulisporipyrone A and scirpyrone H were characterized from endolichenic strains of *Nodulisporium* sp. [143] and an unknown species belonging to the Sarcosomataceae [144], respectively.

Interestingly, some products were first identified from marine strains of uncommon fungal species; this is the case of the remisporines, from the typical marine fungus *Remispora maritima* [145], monodictyphenone and pestalotiorin, respectively, from algal endophytic strains of *Monodictys putredinis* [146] and *Pestalotiopsis* sp. [147]. Moreover, phomaligol A was previously identified as a product of several fungi of marine origin [148], while the more common tenellic acids were first obtained from the freshwater fungus *Dendrospora tenella* [149].

This brief overview on the occurrence of the secondary metabolites of marine-derived *Talaromyces* as products of other fungal species underlines a remarkable biochemical affinity with both *Penicillium* and *Aspergillus*, which can be easily explained in terms of the phylogenetic proximity among these genera. However, their ability to synthesize many products, which are known in more phylogenetically distant fungi, is also quite evident. Although secondary metabolites can be synthesized through various and diverse biochemical pathways in different organisms, the hypothesis of a horizontal transfer of gene clusters encoding for the synthesis of the bioactive secondary metabolites among fungi, which was advanced at the end of the past millennium [150,151], has recently become more and more credited as a process driving the evolution in these organisms. It is also thought to involve their symbiotic associates [152,153], which provides an additional account on the extent of the chemodiversity in fungi characterized by a propensity toward an endophytic/endozoic lifestyle, such as *Talaromyces* [3,4]. In this respect, it is quite amazing to find that the incisterols, reported as products of *T. versatilis* [89], were first identified as a new sterol class from marine sponges [154]. The identification of the new withanolide compound talasteroid [84] is also meaningful, since it follows the finding of withanolide as a secondary metabolite of a strain of *T. pinophilus* endophytic in *Withania somnifera* [155]; notably, withanolides were previously known from plants only, with some products having been reported to possess antifungal activity [156].

## 4. Biological Properties

The research instances supporting the biological characterization of marine-derived *Talaromyces* strains are various. Some strains have shown effectiveness as biocontrol agents against plant pathogens [71,157]; others have been considered as a source of enzymes, such as phytase [62], chitinases, cellulases, and β-glucosidases [66,158], or have been investigated in preliminary assays as a source of pigments [35] and bioactive peptides [159].

In some cases, bioactivity assessments have been carried out at a preliminary stage by using organic extracts without performing product purification, with reference to antioxidant, antitumor, antifungal, antibacterial, acetylcholinesterase, and α-glucosidase inhibitory properties [39,41,43,55,69,85,160]. However, most of the reports in the literature concern the biological properties of purified compounds, as summarized in Table 4. Overall, the available data are indicative of quite variable effects in both qualitative and quantitative terms; however, for the time being, the preliminary nature of many of these studies does not allow for a determination of the applicative relevance of these findings. Indeed, the definition of exhaustive protocols, considering the most accurate assays and most responsive microbial/cell line panels, would help in obtaining a more reliable appreciation of the real potential of these products.

Most of the assays concerning these new compounds were carried out on the inhibitory effects against microbes and cancer cell lines, representing only preliminary indications of their antibiotic and/or antitumor properties. Indeed, more accurate assessments and an elucidation of the mechanisms of action are required for the aim of bringing the best products to the attention of pharmacologists. However, there are some exceptions where bioactivity has been explored with reference to specific targets. This is the case of talaverrucin A, which has been characterized as an inhibitor of the Wnt/β-catenin pathway acting upstream of the β-catenin level [81]. This pathway is known to play a pivotal role in the embryonic development and homeostasis maintenance in vertebrates, and its dysregulation is associated with various diseases, such as congenital malformations and several kinds of cancers [161].

Besides the new findings reported in the recent literature, the biological properties of many of these compounds have been investigated in previous studies, with some of them being characterized as candidate pharmaceutical products. This is the case of bacillisporins, duclauxins, and other oxaphenalenone analogs, with reference to their notable antibacterial and antitumor properties [114]. Antitumor activity has been also documented for 3-*O*-methylfunicone, on account of its multiple concurrent antiproliferative, proapoptotic, and gene-modulatory effects in several tumor cell lines [162,163,164,165,166,167], along with its recently disclosed anticholesterolemic [19] and antiviral properties [168,169]. More generally, these valuable bioactivities have been found to characterize other funicone and vermistatin compounds [112,113]. Many other products deserve consideration for their valuable antitumor and antimicrobial properties, such as depsidones, naphthoquinones, cyclopeptides, and other bioactive peptides, which are quite commonly reported from marine-derived fungi [170,171,172].

The biotechnological exploitation of marine-derived *Talaromyces* products may go well beyond the pharmaceutical field. In fact, besides the antiviral and tyrosine phosphatase inhibitory properties reported in Table 4, chrodrimanins were previously characterized as potent and selective blockers of the *γ*-aminobutyric acid-gated chloride channels in silkworms (*Bombyx mori*), introducing them as a lead for the development of safer pesticides [173]. In this respect, the many isocoumarins have disclosed anti-acetylcholinesterase properties, making them credited for this application, in addition to their possible employment in the treatment of Alzheimer’s disease, as well as other medical disorders, based on their anti-inflammatory and α-glucosidase inhibitory properties [174].

## 5. Conclusions

This overview, considering papers published in the last seven years, resulted in the impressive number of 514 secondary metabolites being extracted from cultures of marine-derived *Talaromyces* strains, depicting the outstanding chemodiversity of these fungi. This conspicuous biochemical booty was derived from investigations on the biosynthetic capacities concerning just 54 strains out of a total of 95 reported from marine sources in this period. Since about 45% of the products were originally identified from these strains, it is reasonable to expect an increase in this number of new compounds as long as the exploration of such a valuable trove is carried on by the scientific community in the future. At the same time, the remarkable proportion of products displaying various kinds of bioactivity introduces perspectives for the identification and possible exploitation of new drug prospects. The extent to which this expectation will materialize is largely dependent on the set up of conventional guidelines for defining effective screening protocols that may enable the performance of more exhaustive assessments of these bioactive properties.

## Figures and Tables

**Figure 1 biomolecules-13-01021-f001:**
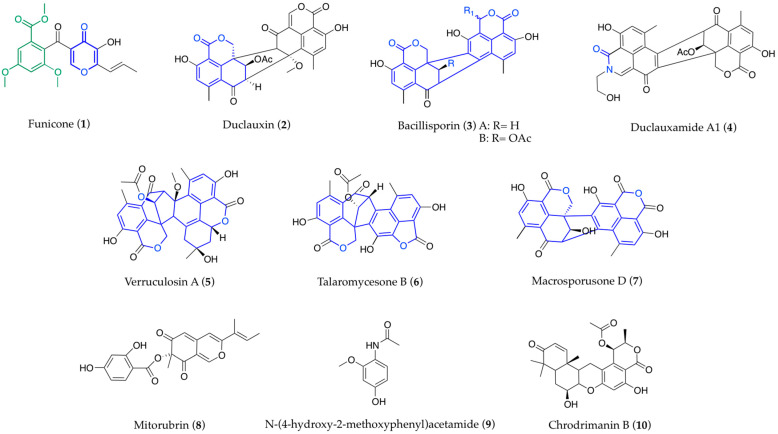
Structures of typical *Talaromyces* secondary metabolites.

**Figure 2 biomolecules-13-01021-f002:**
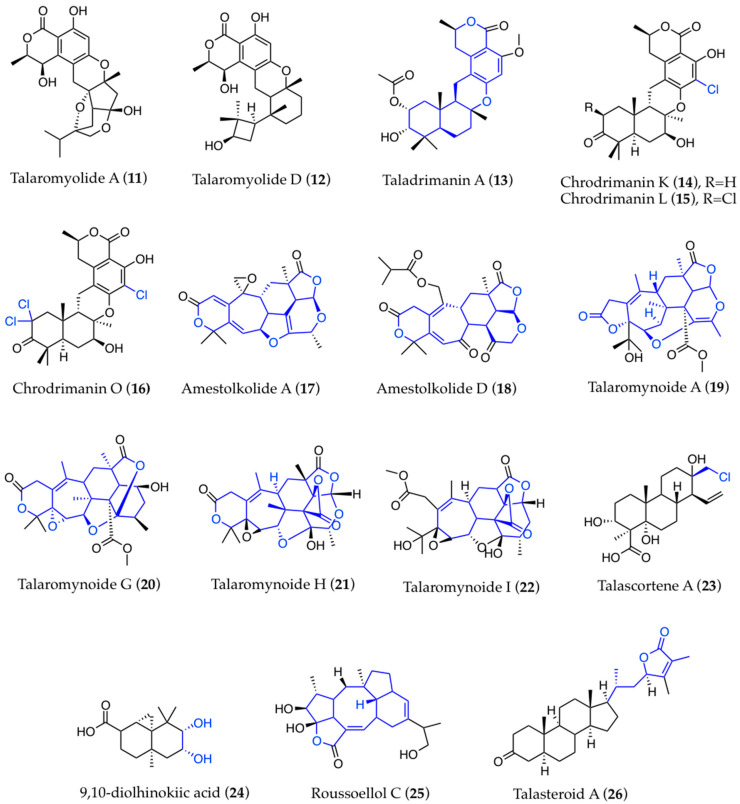
Novel structural models found in meroterpenoids of marine-derived *Talaromyces*.

**Figure 3 biomolecules-13-01021-f003:**
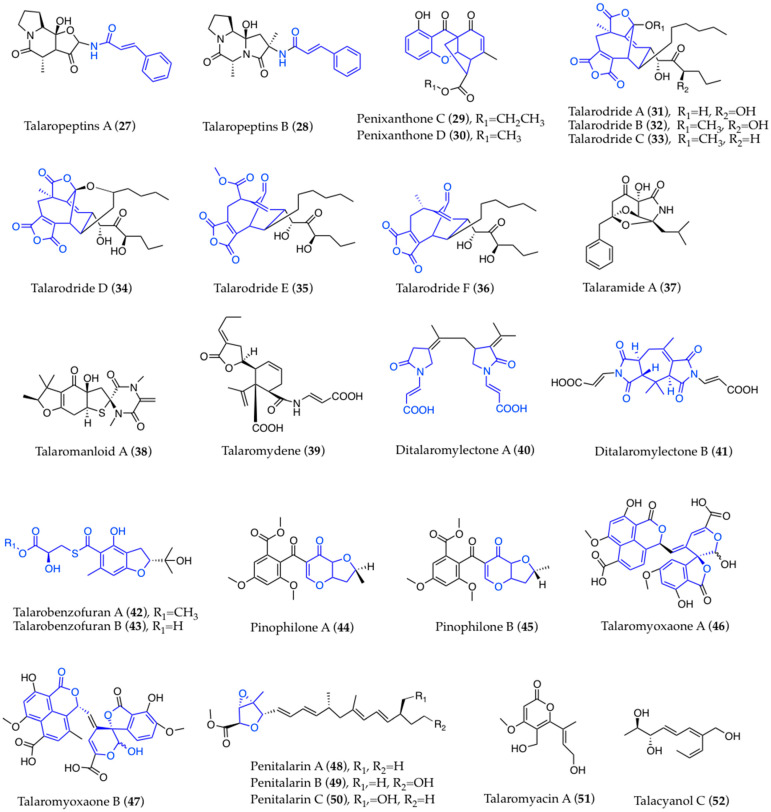
Novel structural models found in secondary metabolites of marine-derived *Talaromyces*.

**Figure 4 biomolecules-13-01021-f004:**
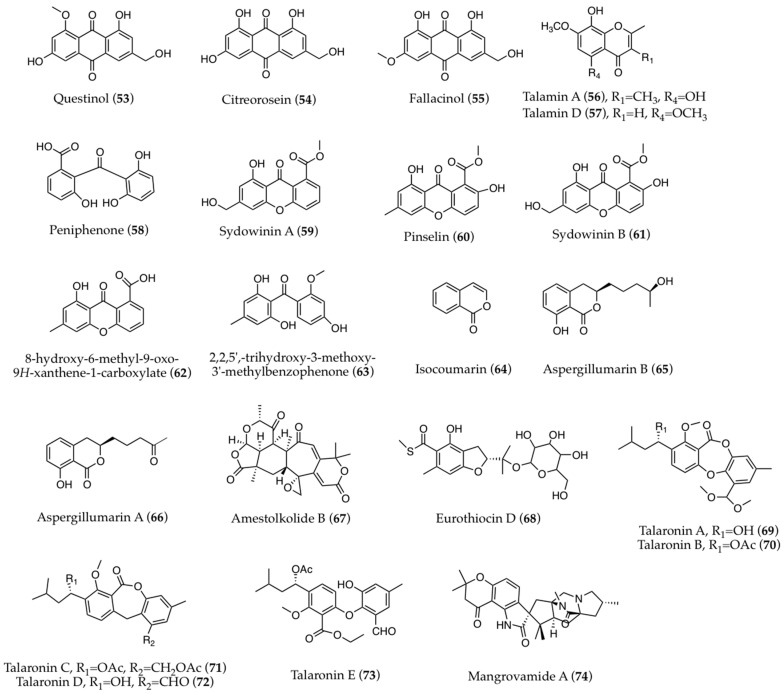
Secondary metabolites of marine-derived *Talaromyces* considered in studies on structure-activity relationships.

**Table 1 biomolecules-13-01021-t001:** *Talaromyces* species reported in the literature from marine sources since 2016.

Species	Source	Location	Reference
*T. aculeatus*	mangrove (*Kandelia candel*, leaf)	Guangdong (China)	[20] *
	deep sea sediment	Indian Ocean	[24] *
	red alga (*Laurencia obtusa*)	Suez Gulf (Egypt)	[29] *
*T. albobiverticillius*	coral rubble, sediment	La Reunion Island	[31] *
	unidentified ascidian	Manado (Indonesia)	[22] *
*T. amestolkiae*	mangrove (*Kandelia obovata*, leaf)	Guangdong (China)	[32] *
	pipefish (Syngnathus acus) †	Hainan (China)	[33] *
*T. assiutensis*	mangrove (*Ceriops tagal*, leaf)	South China Sea	[34] *
	mangrove (*Avicennia marina*, root)	Maharashtra (India)	[35]
*T. brunneus*	sponge (*Axinella polypoides*)	Marmara (Turkey)	[36]
*T. cellulolyticus*	coral	South China Sea	[37] *
*T. cyanescens*	green alga (*Caulerpa* sp.)	Da Nang (Vietnam)	[38] *
*T. flavus*	sediment	Kanyakumari district (India)	[39]
	mangrove (*Acanthus ilicifolius*, stem)	Hainan (China)	[40] *
	sponge (*Mycale* sp.)	Samaesarn Island (Thailand)	[41]
*T. funiculosus*	deep sea sediment	Shimokita Peninsula (Japan)	[42]
	mangrove sediment	Hainan (China)	[23,25] *
	coral (*Porites compressa*)	Zhanjiang (China)	[43]
	sea cucumber (*Holothuria leucospilota*)	Pangkor Island (Malaysia)	[44]
	deep sea sediment	South China Sea	[45] *
*T. fuscoviridis*	mangrove rhizosphere	Hainan (China)	[46]
*T. haitouensis*	mudflat in estuary	Jiangsu (China)	[16]
*T. helicus*	deep sea sediment	South China Sea	[47] *
*T. indigoticus*	deep sea sediment	South China Sea	[48] *
*T. islandicus*	red alga (*Laurencia okamurai*)	Qingdao (China)	[49] *
*T. liani*	mudflat in intertidal zone	Yongyudo (South Korea)	[50]
*T. mangshanicus*	sediment	South China Sea	[51] *
*T. minioluteus*	sediment	East China Sea	[27] *
	mussel (*Gigantidas platifrons*)	South China Sea	[52] *
*T. pinophilus*	mangrove sediment	Xiamen (China)	[19] *
	mangrove rhizosphere	Techeng Isle (China)	[26] *
	mangrove (*A. marina*) rhizosphere	Gazi Bay (Kenya)	[53]
	sponge (*Mycale* sp.)	Samaesarn Island (Thailand)	[54] *
*T. purpureogenus*	mudflat in intertidal zone	Tianjin (China)	[18] *
	brown alga	Kovalam (India)	[55]
	mud at the coastline	Hebei (China)	[56] *
	brown alga (*Phaeurus antarcticus*)	Half Moon Island (Antarctica)	[57]
	red alga (*Grateloupia filicina*)	Zhejiang (China)	[58] *
	soft coral	Nansha islands (China)	[59] *
	soft coral	South China Sea	[60] *
	brown alga (*Sargassum muticum*)	Kerala (India)	[61]
	water	Sharm El-Sheikh governorate (Egypt)	[62]
*T. rotundus*	reef water	La Reunion Island	[31]
*T. rugulosus*	sponge (*Axinella cannabina*)	Sığaçık-İzmir (Turkey)	[63] *
*T. scorteus*	sea anemone (*Cerianthus* sp.)	Magellan Sea Mounts	[64] *
*Talaromyces* sp.	mangrove (*Sonneratia apetala*, leaf)	Guangdong (China)	[17] *
	unidentified tunicate	Tweed Heads (Australia)	[65] *
	annellid (*Sipunculus nudus*)	Haikou Bay (China)	[21] *
	mangrove (*Rhizophora mucronata*, root)	Andaman Islands (India)	[66]
	mangrove (*Laguncularia racemosa*) rhizosphere	Vera Cruz (Mexico)	[67]
	abandoned saltern mudflat in intertidal zone	Yubudo (South Korea) Gopado, Yongyudo (South Korea)	[50]
	mangrove (*R. mucronata*) rhizosphere	Gazi Bay, Mida Creek (Kenya)	[53]
	coral (*Porites pukoensis*)	Zhanjiang (China)	[43]
	mangrove (*K. obovata*, fruit)	Guangxi (China)	[68] *
	mangrove (*Brownlowia tersa*, stem)	Sundarbans (Bangladesh)	[69]
	mangrove (*Ceriops decandra*, bark)		
	mangrove (*Heritiera fomes*, bark)		
	mangrove (*Xylocarpus granatum*, bark)		
	mangrove (*Xylocarpus moluccensis*, bark)		
	halibut (*Hippoglossus* sp.)	Zhejiang (China)	[70] *
	water	Yap Trench	[71]
	sponge	Weddell Sea (Antarctica)	[72] *
	mangrove (*X. granatum*, root)	Hainan (China)	[30] *
	sediment	Zhejiang (China)	[73] *
	mudflat in intertidal zone	Qingdao (China)	[74] *
	mangrove (*Kandelia* sp., leaf)	Guangdong (China)	[75] *
	mangrove soil	Hainan (China)	[76,77] *
	water	Dongshan Island (China)	[78,79] *
	unidentified sponge	Bulon Island (Thailand)	[80] *
	unidentified sponge	Prydz Bay (Antarctica)	[81] *
*T. stipitatus*	mangrove (*A. ilicifolius*, leaf)	Guanxi (China)	[82] *
	sponge (*Stylissa flabelliformis*)	Samaesarn Island (Thailand)	[83] *
	mangrove (*A. marina*, root)	Tamil Nadu (India)	[66]
	mudflat in intertidal zone	Yongyudo (South Korea)	[50]
	mangrove (*R. mucronata*) rhizosphere	Gazi Bay (Kenya)	[53]
	brown alga (*S. muticum*)	Kerala (India)	[61]
*T. stollii*	unknown	Bohai Sea (China)	[84] *
*T. trachyspermus*	sponge (*Clathria reinwardti*)	Kram Island (Thailand)	[85]
*T. tratensis*	sponge (*Mycale* sp.)	Samaesarn Island (Thailand)	[86] *
*T. variabilis*	mangrove rhizosphere	Fujian (China)	[24] *
*T. verruculosus*	reef water	La Reunion Island	[31]
	soft coral (*Goniopora* sp.)	Hainan (China)	[87] *
	mangrove (*A. marina*) rhizosphere	Mida Creek (Kenya)	[53]
	mangrove (*C. tagal*) rhizosphere	Gazi Bay (Kenya)	
	mangrove (*X. moluccensis*, pneumatophores)	Sundarbans (Bangladesh)	[69]
	deep-sea sediment	Okinawa Trough	[88]
	mangrove (*X. granatum*)	South China Sea	[28] *
*T. versatilis*	soft coral	Yongxing Island (China)	[89] *
*T. zhenhaiensis*	mudflat in estuary	Zhejiang (China)	[16]

† This isolation source is unreliable, considering that the authors describe it as a “marine herb”; * these entries report on strains used for identification of secondary metabolites.

**Table 2 biomolecules-13-01021-t002:** Novel compounds reported as secondary metabolites of marine-derived *Talaromyces*.

Compound	Species	Reference
** *Alkaloids* **		
Chaetominine B	*T. helicus*	[47]
Ditalaromylectones A–B	*T. mangshanicus*	[51]
11,17-*epi*-Mangrovamide A	*T. funiculosus*	[43]
Mangrovamides D–K	*T. funiculosus*	[25]
Mangrovlide A	*T. funiculosus*	[90]
Talaramide A	*T. amestolkiae*	[91]
Talaromanloid A	*T. mangshanicus*	[51]
** *Amides* **		
Penicimumide	*T. purpureogenus*	[56]
Talaromydene	*T. mangshanicus*	[51]
Talaromydien A	*Talaromyces* sp. *T. verruculosus*	[70] [92]
Talaromylectone	*T. mangshanicus*	[51]
** *Anthraquinones* **		
4-8-Dihydroxyconiothyrinone B, 8-11-dihydroxyconiothyrinone B, 8-hydroxyconiothyrinone B, 8-dihydroxy-10-*O*-methyldendryol E	*T. islandicus*	[93]
2,2′-*bis*-(7-Methyl-1,4,5-trihydroxy-anthracene-9,10-dione)	*T. stipitatus*	[83]
Rugulosin D	*Talaromyces* sp.	[74]
** *Azaphilones* **		
Azaphilone compounds 1–3 and 5	*T. indigoticus*	[94]
7-*epi*-Pinazaphilone B	*T. pinophilus*	[54]
Talaromyacins A–C	*T. purpureogenus*	[95]
** *Benzofurans* **		
Eurothiocins C–H	*T. indigoticus*	[48]
1-(5-Hydroxy-7-methoxybenzofuran-3-yl)ethan-1-one, 5-hydroxy-7-methoxy-2-methylbenzofuran-3-carboxylic acid	*T. amestolkiae*	[32]
(2-Hydroxypropan-2-yl)-6-methyl-2,3-dihydrobenzofuran-4-ol	*T. indigoticus*	[96]
Isoprenyl-benzofuran derivative	*T. indigoticus*	[94]
Talabenzofurans A–C	*Talaromyces* sp.	[76]
Talarominine A	*T. minioluteus*	[52]
** *Benzophenones* **		
2,2′,5′-Trihydroxy-3-methoxy-3′-methylbenzophenone, 2,2′,3,5-tetrahydroxy-3′-methylbenzophenone	*T. islandicus*	[49]
** *Benzoquinones* **		
Anserinone C	*Talaromyces* sp.	[79]
** *Chromones* **		
2-(2′-Hydroxypropyl)-5-methyl-7,8-dihydroxychromone	*T. aculeatus*	[20]
** *Decalins* **		
Fusarielins O–P	*Talaromyces* sp.	[73]
** *Depsidones* **		
5′-Hydroxypenicillide	*T. pinophilus*	[19]
Talamins A–D	*T. minioluteus*	[52]
Talaromyones A–B	*T. stipitatus* *Talaromyces* sp.	[82] [77]
Talaronins A–H	*Talaromyces* sp.	[77]
** *Diphenyl ethers* **		
2-Hydroxy-6-(2′-hydroxy-3′-hydroxymethyl-5-methylphenoxy)-benzoic acid	*T. albobiverticillius*	[22]
** *Funicones* **		
Pinophilones A–E	*T. pinophilus*	[26]
** *Furans* **		
Talarofuranone, talarotetrahydrofuran	*Talaromyces* sp.	[80]
** *Indenes* **		
1,2-Indandiol	*T. funiculosus*	[23]
** *Isocoumarins* **		
Aspergillumarin C	*Talaromycs* sp.	[75]
5,6-Dihydroxy-3-(4-hydroxypentyl)-isochroman-1-one, 6,8-dihydroxy-5-methoxy-3-methyl-isochromen-1-one, 5-hydroxy-4-(1-hydroxyethyl)-8-methoxyisocoumarin, 6-hydroxy-8-methoxy-3,4-dimethylisocoumarin, isobutyric acid 5,7-dihydroxy-2-methyl-4-oxo-3,4-dihydro-naphththalen-1-yl methyl ester	*T. amestolkiae*	[32]
3-(4,5-Dihydroxy-pentyl)-8-hydroxy-isochroman-1-one	*T. amestolkiae* *T. flavus*	[32] [40]
Penicimarins L–M	*Talaromyces* sp.	[30]
Peniisocoumarin H	*T. minioluteus*	[27]
Talaroisocoumarin A	*Talaromyces* sp.	[70]
Talaromarins A–F	*T. flavus*	[40]
Talumarins A–B	*T. rugulosus*	[63]
Tratenopyrone	*T. tratensis*	[86]
** *Ketones* **		
6-(2-Carboxyvinyl)-N-GABA-PP-V	*T. albobiverticillius*	[97]
Penicillquei C	*T. verruculosus*	[92]
Penicimutanolones A–B, penicimutanolone A methyl ether	*T. purpureogenus*	[56]
Penitalarins A–C	*T. aculeatus/T. variabilis*	[24]
2-Prop-1-en-1-yl-oct-4-ene-1,6,7-triol	*T. indigoticus*	[96]
Purpurofuranone	*T. purpureogenus*	[98]
Purpurogenic acid	*T. purpureogenus*	[99]
Purpuropyranone	*T. purpureogenus*	[98]
Talarocyclopenta A–C	*T. assiutensis*	[34]
** *Lactones* **		
Lactone acid *n*-butyl ester, lactone diacid 7-*O*-*n*-butyl ester, 4-methoxylactone acid *n*-butyl ester	*T. rugulosus*	[40]
5-Methylhexahydrofuro[2,3-*b*]furan-2-yl-ethanol	*T. indigoticus*	[96]
Nafuredin B	*T. aculeatus/T. variabilis*	[24]
*cis*-Resorcylide, 7-*O*-*n*-butylresorcylides, 7-hydroxyresorcylides, 7-methoxyresorcylides	*T. rugulosus*	[40]
Talarodilactones A–B	*T. rugulosus*	[40]
** *Meroterpenoids* **		
Amestolkolides A–D	*T. amestolkiae*	[100]
Chrodrimanins K–S	*Talaromyces* sp.	[21,101]
Chromosulfine	*T. purpureogenus*	[102]
Taladrimanin A	*Talaromyces* sp.	[78]
Talaromynoids A–I	*T. purpureogenus*	[59]
Talaromyolides A–K	*T. purpureogenus*	[58,103]
Talaromytin	*T. purpureogenus*	[58]
** *Morpholinones* **		
Talaromorpholinone	*Talaromyces* sp.	[80]
** *Naphthoquinones* **		
Talanaphthoquinones A–B	*Talaromyces* sp.	[68]
** *Nonadrides* **		
Talarodrides A–F	*Talaromyces* sp.	[72]
** *Phenalenones* **		
Abeopyrenulin, 11-apopyrenulin	*T. purpureogenus*	[60]
Amestolkins A–B	*T. amestolkiae*	[33]
Bacillisporins K–L	*Talaromyces* sp.	[74]
Dihydroxy-ergosta-4,6,8(14)-tetraen-3-one	*T. pinophilus*	[54]
Penicimutalidine	*T. purpureogenus*	[104]
Penicimutamides A–E	*T. purpureogenus*	[105,106]
Penicimutanin C	*T. purpureogenus*	[107]
Talaromyoxaones A–B	*T. purpureogenus*	[60]
Talaropinophilide, talaropinophilone	*T. pinophilus*	[54]
Talaverrucin A	*Talaromyces* sp.	[81]
Verruculosins A–B	*T. verruculosus*	[87]
** *Peptides* **		
Penicimutide	T*. purpureogenus*	[18]
Talaropeptins A–B	*T. purpureogenus*	[108]
** *Polyenes* **		
Talacyanols A–C	*T. cyanescens*	[38]
** *Polyphenols* **		
Talaversatilis A–B	*T. versatilis*	[89]
** *Pyrones* **		
Talapyrones A–B	*Talaromyces* sp.	[76]
** *Pyrroles* **		
(*R*)-3-Hydroxy-2,7-dimethylfuro[3,4-*b*]pyridin-5(7*H*)-one	*Talaromyces* sp.	[79]
10-Hydroxy-8-demethyltalaromydine, 11-hydroxy-8-demethyltalaromydine	*T. mangshanicus*	[51]
** *Sterols* **		
Cyclosecosteroid A	*Talaromyces* sp.	[75]
Talarosterone	*T. stipitatus*	[83]
Talasteroid	*T. stollii*	[84]
** *Sulfones* **		
Pensulfonamide, pensulfonoxy	*T. aculeatus*	[29]
** *Terpenes* **		
Dihydroxyisocupressic acid	*T. scorteus*	[64]
9,10-Diolhinokiic acid	*T. purpureogenus*	[109]
Purpurides E–G	*T. minioluteus*	[27]
Roussellol C	*T. purpureogenus*	[109]
Talascortenes A–G	*T. scorteus*	[64]
Verruculides B_2_–B_3_	*Talaromyces* sp.	[21]
** *Xanthones* **		
Penixanthones A–D	*T. funiculosus*	[23,90]
1,4,7-Trihydroxy-6-methylxanthone	*T. islandicus*	[49]

**Table 3 biomolecules-13-01021-t003:** Secondary metabolites identified as products of marine-derived *Talaromyces* that are also known from other biological sources.

Compound	Species	Reference
** *Acids* **		
Asperitaconic acid B, butylitaconic acid	*T. assiutensis*	[34]
Bromothiobenzoic acid	*T. aculeatus*	[29]
Coculnol, acetylcoculnol	*Talaromycs* sp.	[79]
Hydroxybenzoic acid	*T. versatilis*	[89]
8-Hydroxy-carboxy-methylenenonanoic acid, 9-hydroxy-carboxy-methylenenonanoic acid	*T. assiutensis*	[34]
Isocyclopaldic acid	*T. funiculosus*	[45]
Methylcurvulinate	*T. minioluteus*	[27]
Methylorsellinate	*T. indigoticus*	[96]
** *Alcohols* **		
*bis*-Methoxybenzyl-butanediol	*T. tratensis*	[86]
** *Alkaloids* **		
Alantrypinone	*T. verruculosus*	[28]
Chaetominine	*T. helicus*	[47]
Cyclotryprostatin B	*T. helicus*	[47]
Cyclotryprostatin E	*T. purpureogenus*	[109]
Dihydroxyfumitremorgin C	*T. helicus*	[47]
Fructigenines A–B	*T. purpureogenus*	[107]
Fumigaclavine C, fumigatin oxide, fumiquinazolines F, G, J	*T. helicus*	[47]
Mangrovamides A, C, G, I	*T. funiculosus*	[25,45]
Methoxyspirotryprostatin B	*T. purpureogenus* *T. helicus*	[109] [47]
Methyl-hexahydro-pyrazino-pyrido-indole-dione	*T. purpureogenus*	[92]
Penicimutanin A	*T. purpureogenus*	[107]
Premalbrancheamide	*T. purpureogenus*	[106]
Pseurotin A, F1, methylpseurotin A, norpseurotin A	*T. helicus*	[47]
Rugulosuvine A	*T. purpureogenus*	[107]
Spiro-dipyrrolo-pyrazine-indole-trione	*T. helicus*	[47]
Tryptoquivalines F, J, isotryptoquivaline F	*T. helicus*	[47]
* **Amides** *		
Hydroxy-methoxyphenyl-acetamide	*T. cellulolyticus*	[37]
Hydroxy-methyl-oxobutyl-butanamide	*Talaromyces* sp.	[80]
** *Anthraquinones* **		
Acetylquestinol	*T. pinophilus*	[34]
Citrorosein	*T. stipitatus* *T. minioluteus*	[83] [27]
Dihydroxy-methoxy-methyl-anthracene-dione	*T. funiculosus*	[23]
Emodin, fallacinol, questinol, rheoemodin	*T. stipitatus*	[83]
Questin	*T. funiculosus*	[45]
Rugulosin A	*Talaromyces* sp.	[74]
** *Azaphilones* **		
FK17-P2b1	*T. minioluteus* *Talaromyces* sp.	[27] [78]
Glutarylmonascorubraminic acid, hydroxyethyl-monascorubramin, threonine-monascorubramine, threonine-rubropunctamine, GABA-rubropunctatin	*T. albobiverticillius*	[110]
Mitorubrin	*T. purpureogenus* *Talaromyces* sp.	[99] [73]
Mitorubrinol	*T. purpureogenus*	[99]
Monascorubramine, glutarylrubropunctamine, glycylrubropunctatin	*T. albobiverticillius*	[111]
Peniazaphilin B	*Talaromyces* sp.	[76,79]
Pinazaphilone B	*Talaromyces* sp.	[73]
Pinophilin	*Talaromyces* sp.	[73]
Pinophilins B, G	*T. pinophilus*	[26]
Purpurquinone A	*T. minioluteus*	[27]
Sch1385568	*T. pinophilus*	[34]
Sch725680	*T. pinophilus*	[26]
Sequoiamonascin C	*Talaromyces* sp.	[73]
Wortmin	*T. tratensis*	[86]
** *Benzaldehydes* **		
Dihydroxybenzaldehyde	*Talaromyces* sp.	[77]
Ethyl-dihydroxy-methylbenzaldehyde	*Talaromyces* sp.	[78]
Hydroxybenzaldehyde, hydroxy-methylbutenyl-benzaldehyde	*Talaromyces* sp.	[79]
** *Benzofurans* **		
Carboxy-methyl-butenyl-octahydro-methoxycarbonyl-3-methyl-methylene-oxo-benzofuranacetic acid	*Talaromyces* sp.	[80]
Dihydroxy-dimethyl-dibenzofuran	*T. versatilis*	[89]
Eurothiocin A	*T. cyanescens* *T. indigoticus* *Talaromyces* sp.	[38] [48] [76]
Purpuresters A–B	*T. minioluteus*	[27]
Trypacidin	*T. helicus*	[47]
** *Benzoquinones* **		
Fumiquinone B	*T. helicus*	[47]
** *Cerebrosides* **		
Flavuside B	*T. verruculosus*	[28]
** *Cyclopentenones, Cyclohexenones* **		
Phomaligol A	*T. funiculosus*	[45]
Terrein	*T. verruculosus*	[92]
** *Decalins* **		
Fusarielin M	*Talaromyces* sp.	[73]
** *Depsidones* **		
Dehydroisopenicillide, dehydroxypenicillide, purpactin C	*T. pinophilus*	[19]
Isopenicillide	*T. pinophilus*	[19,26]
Methyldehydroisopenicillide	*T. pinophilus*	[26]
Penicillide	*T. pinophilus* *T. funiculosus* *T. stipitatus* *T. verruculosus* *Talaromyces* sp.	[19,26] [23] [82] [28] [74]
Purpactin A (=vermixocin B)	*T. pinophilus* *T. stipitatus* *Talaromyces* sp.	[19] [82] [77]
Purpactin C’	*Talaromyces* sp.	[77]
Secopenicillide A	*T. pinophilus* *Talaromyces* sp.	[19] [77]
Secopenicillide B	*T. stipitatus* *Talaromyces* sp.	[82] [77]
** *Diphenyl ethers* **		
Diorcinol, methyldiorcinol, methoxycarbonyldiorcinol	*T. versatilis*	[89]
Methoxy-methyl-biphenyltriol	*T. mangshanicus*	[51]
Methyl tenellate	*T. pinophilus*	[19]
Tenellic acid A	*T. stipitatus*	[82]
Tenellic acid C	*T. stipitatus* *Talaromyces* sp.	[82] [77]
** *Esters* **		
Ethyl everninate	*T. indigoticus*	[96]
Methyl-hydroxy-methylhexenoate, methyl-hydroxyphenyl-acetate	*T. minioluteus*	[27]
** *Funicones* **		
Demethylvermistatin, *epi*-hydroxydihydrovermistatin, methyldihydrovermistatin, penisimplicissin, demethylpenisimplicissin, penicidones C–D	*T. pinophilus*	[26]
Dihydrovermistatin	*T. pinophilus* *Talaromyces* sp.	[26] [78]
Funicone, deoxyfunicone	*T. pinophilus*	[19]
Methylfunicone, hydroxyvermistatin, methoxyvermistatin	*T. pinophilus*	[19,26]
Vermistatin	*T. pinophilus* *Talaromyces* sp.	[19,26] [73,78]
** *Furans* **		
Azaspirofuran A	*T. helicus*	[47]
Cillifuranone	*T. purpureogenus*	[98]
** *Glycosides* **		
Carnemycins B, E	*T. verruculosus*	[28]
** *Isocoumarins* **		
Aspergillumarin A	*T. amestolkiae* *T. flavus* *T. rugulosus* *T. verruculosus* *Talaromyces* sp.	[32] [40] [63] [92] [30,70,75]
Aspergillumarin B	*T. amestolkiae* *T. verruculosus* *Talaromyces* sp.	[32] [92] [70,75]
Dihydroxy-2-hydroxypropyl-methylisochromenone, dihydroxy-2*S*-hydroxypropyl-methylisochromenone	*T. flavus* *Talaromyces* sp.	[40] [78]
Dihydroxyl-oxoisochromanyl-propanoic acid	*Talaromyces* sp.	[75]
Dihydroxymellein	*Talaromyces* sp.	[77]
Dihydroxy-trimethylisochromanone, dihydroxy-trimethylisochroman	*Talaromyces* sp.	[79]
Dimethyl-dihydroxyisocoumarin	*T. amestolkiae*	[32]
Hydroxy-hydroxymethyl-methoxy-methylisocoumarin	*T. amestolkiae*	[32]
Hydroxy-hydroxypropyl-methoxyisochromanone	*T. flavus*	[40]
Hydroxymellein	*T. cellulolyticus* *Talaromyces* sp.	[37] [78,79]
Hydroxy-methoxy-dimethylchromone	*T. minioluteus*	[52]
Hydroxy-methoxy-methylphthalide	*T. funiculosus*	[90]
Hydroxy-methyl-dimethoxycoumarin	*Talaromyces* sp.	[70]
Hydroxypropyl-hydroxy-dihydroisocoumarin	*T. flavus*	[40]
Hydroxyramulosin	*Talaromyces* sp.	[76]
Orthosporin	*T. minioluteus*	[27]
Penicifuran A	*Talaromyces* sp.	[70]
Peniciisocoumarins A–G	*T. flavus*	[40]
Peniciisocoumarin D	*Talaromyces* sp.	[70]
Peniciisocoumarins E–F	*Talaromyces* sp.	[30]
Penicilloxalone B	*Talaromyces* sp.	[30,70]
Penicimarin B	*T. amestolkiae*	[32]
Penicimarin C	*T. amestolkiae* *T. flavus*	[32] [40]
Penicimarin G	*T. flavus* *Talaromyces* sp. *T. verruculosus*	[40] [30] [92]
Penicimarin H	*T. flavus* *Talaromyces* sp.	[40] [30]
Penicimarin I	*Talaromyces* sp.	[30]
Penicimarin N	*T. flavus*	[40]
Pestalotiorin	*T. flavus* *Talaromycs* sp.	[40] [79]
Ramulosin	*T. cyanescens* *Talaromyces* sp.	[38] [76]
Sclerotinin A	*Talaromyces* sp.	[78,79]
Sclerotinin B	*Talaromyces* sp.	[79]
Sescandelin	*T. amestolkiae* *Talaromyces* sp.	[32] [70]
Sescandelin B	*T. amestolkiae*	[32]
Trihydroxy-hydroxyethylisocoumarin	*T. amestolkiae* *Talaromyces* sp.	[32] [70]
** *Ketones* **		
Dihydro-hydroxy-hydroxymethyl-methoxy-methylnaphtho-furandione	*Talaromyces* sp.	[68]
Methyl-dihydropyranone	*Talaromyces* sp.	[78]
Penicillquei A	*T. verruculosus*	[92]
** *Lactones* **		
Aspergilactone B	*T. verruculosus*	[92]
Carboxyphthalide	*T. aculeatus*	[20]
Corymbiferan lactone A	*T. purpureogenus*	[104]
Dehydromevalonic lactone, mevalonolactone	*T. funiculosus*	[90]
Deoxyrubralactone	*T. pinophilus*	[34]
Lactone acid, lactone diacid	*T. rugulosus*	[63]
Nafuredin A	*T. aculeatus/T. variabilis* *T. mangshanicus*	[24] [51]
** *Meroterpenoids* **		
Austinolide	*T. purpureogenus* *T. mangshanicus* *T. stollii*	[103] [51] [84]
Austin, austinol, dehydroaustin	*T. stollii*	[84]
Berkeleyacetal, berkeleyacetal A, epoxyberkeleydione	*T. purpureogenus*	[59]
Chrodrimanins A–B	*T. amestolkiae* *Talaromyces* sp. *T. cellulolyticus* *T. stollii*	[100] [21,78] [37] [84]
Chrodrimanin C	*T. cellulolyticus* *T. stollii*	[37] [84]
Chrodrimanin E	*Talaromyces* sp.	[101]
Chrodrimanin F	*Talaromyces* sp. *T. cellulolyticus*	[101] [37]
Chrodrimanin H	*Talaromyces* sp. *T. cellulolyticus*	[21,78] [37]
Dehydroaustinol	*T. mangshanicus* *T. stollii*	[51] [84]
Hydroxypentacecilide A	*Talaromyces* sp.	[101]
Miniolutelide C	*T. purpureogenus*	[59]
Preaustinoid	*T. purpureogenus*	[103]
Purpurogenolide E	*T. amestolkiae*	[100]
Territrem B	*T. verruculosus*	[92]
Thailandolide B	*Talaromyces* sp.	[79]
Verruculide A	*T. cellulolyticus*	[37]
Verruculide B	*Talaromyces* sp.	[101]
** *Naphthoquinones* **		
Acetonyl-methyl-hydroxy-methoxy-naphthazarin, acetyloxyethyl-hydroxy-dimethoxy-naphthalenedione, hydroxy-hydroxyethyl-dimethoxy-naphthalenedione	*Talaromyces* sp.	[68]
Anhydrofusarubin	*Talaromyces* sp.	[68]
Ethyl-dimethoxyjuglone	*Talaromyces* sp.	[68]
Javanicin, anhydrojavanicin	*Talaromyces* sp.	[68]
** *Peptides* **		
Cyclo(l-Val- l-Pro), cyclo(l-Ile- l-Pro), cyclo(l-Leu- l-Pro), cyclo(l-Phe- l-Pro)	*T. purpureogenus*	[18]
** *Phenalenones* **		
Bacillisporin A	*Talaromyces* sp. *T. pinophilus*	[81] [34]
Bacillisporin B	*T. aculeatus* *Talaromyces* sp. *T. pinophilus*	[20] [74] [34]
Bacillisporin C	*T. aculeatus* *T. purpureogenus* *Talaromyces* sp.	[20] [104] [77]
Bacillisporin F	*T. verruculosus*	[87]
Dihydroxy-hydroxybenzylidene-methylbutenyl-indane-carboxylic acid methyl ester	*T. verruculosus*	[28]
Duclauxin, xenoclauxin	*T. verruculosus*	[87]
Macrosporusone D	*Talaromyces* sp.	[74]
SF226	*T. purpureogenus*	[104]
** *Phenols* **		
Acetamidophenol	*Talaromyces* sp.	[70]
Alternaphenol B	*Talaromyces* sp.	[77]
Altenusin	*T. mangshanicus* *Talaromyces* sp.	[51] [73]
Expansols C–F	*T. versatilis*	[89]
Hydroxymethyl-methyl-heptenylphenol	*T. versatilis*	[89]
Methyl-hydroxy-trimethylphenylpropionate	*T. funiculosus*	[90]
Pyrocatechol	*Talaromyces* sp.	[79]
Talaromycin C, deacetyltalaromycin C	*T. pinophilus* *Talaromyces* sp.	[19] [77]
Trihydroxybutyl-hydroxy-hydroxy-methylphenoxy-methylphenylacetate	*T. versatilis*	[89]
Tyrosol	*T. verruculosus*	[28]
** *Phenones* **		
Isomonodictyphenone	*T. versatilis*	[89]
Monodictyphenone	*T. albobiverticillius*	[22]
** *Pyridines* **		
Aminopyridine	*T. verruculosus*	[28]
** *Pyrones* **		
Dihydroaspyrone	*T. indigoticus*	[96]
Fonsecinone A	*T. aculeatus*	[29]
Nodulisporipyrone A	*Talaromyces* sp.	[76]
Scirpyrone H, xylapyrone E	*T. indigoticus*	[96]
Similanpyrone B, hydroxy-dimethylpyrone	*Talaromyces* sp.	[77]
Taiwapyrone	*T. purpureogenus*	[98]
** *Pyrrolidines* **		
Dioxo-propanylidene-pyrrolidinyl- acrylic acid, propanylidene-pyrrolidine-dione	*T. mangshanicus*	[51]
** *Sterols* **		
Cerevisterol	*Talaromycs* sp.	[75]
Cyathisterone	*T. stipitatus*	[83]
Dankasterone	*T. purpureogenus*	[109]
Dankasterone B	*T. funiculosus*	[25]
Epidioxyergostadienol	*T. verruculosus* *Talaromyces* sp.	[28] [75]
Ergostatrienol	*T. aculeatus*	[29]
Ergosterol, ergostadienetetraol, ergostadienetriol	*T. albobiverticillius* *T. verruculosus* *Talaromyces* sp.	[111] [28] [75]
Ergosterol-endoperoxide, ergostatetraenone	*T. stipitatus*	[83]
Ganodermaside A	*T. verruculosus*	[28]
Helvolic acid	*T. aculeatus*	[29]
Hydroxy-ergostatrienone	*T. stollii*	[84]
Methylincisterol, dimethylincisterol A3	*T. versatilis*	[89]
** *Terpenes* **		
Berkedrimane B	*T. minioluteus*	[27]
Hydroxyconfertifolin	*T. minioluteus*	[27]
Penioxalicin	*Talaromyces* sp.	[80]
Piniterpenoid D	*T. pinophilus*	[34]
Sordarin	*Talaromyces* sp.	[65]
Solaniol	*Talaromyces* sp.	[68]
** *Xanthones* **		
Conioxanthone A	*Talaromycs* sp.	[17]
Dihydroxymethyl-hydroxymethylxanthone	*T. funiculosus*	[45]
Leptosphaerin G	*T. funiculosus*	[25]
Pinselin, methyl-hydroxy-methyl-oxo-xanthene-carboxylate, sydowinins A–B	*Talaromycs* sp.	[17]
Remisporine B, *epi*-remisporine B	*Talaromycs* sp.	[17]
Secalonic acid A	*T. stipitatus*	[83]
Secalonic acid D	*Talaromyces* sp.	[77]
Trihydroxymethylxanthone	*T. islandicus*	[49]

**Table 4 biomolecules-13-01021-t004:** Bioactivities of secondary metabolites produced by marine-derived *Talaromyces* strains.

Compound Name	Reported Bioactivities ^1^	References
Acetonyl-methyl-hydroxy-methoxy-naphthazarin, acetyloxyethyl-hydroxy-dimethoxy-naphthalenedione	anti-inflammatory, cytotoxic (RAW 264.7)	[68]
Alantrypinone	α-glucosidase inhibitor	[28]
Altenusin	antioxidant, cytotoxic (B16, MCF-7, HepG2) antibacterial (*S. aureus*), antifungal (*C. albicans*)	[73] [51]
Amestolkolides A–B	anti-inflammatory	[100]
Amestolkines A–B	anti-inflammatory	[33]
Anhydrofusarubin, anhydrojavanicin	anti-inflammatory	[68]
Anserinone C	antibacterial (*S. aureus*), cytotoxic (MKN1)	[79]
Aspergillumarin A	α-glucosidase inhibitor antibacterial (*E. coli*, MRSA), antifungal (*C. albicans*) antioxidant	[32] [70] [30]
Aspergillumarin B	α-glucosidase inhibitor	[32]
Asperitaconic acid B	anti-inflammatory	[34]
Austin, austinol, austinolide, dehydroaustin, dehydroaustinol	antioxidant	[84]
Azaspirofuran A	anti-inflammatory	[47]
Bacillisporin A	α-glucosidase inhibitor, antibacterial (*B. subtilis*) antibacterial (*S. aureus*, MRSA)	[20] [54]
Bacillisporin B	α-glucosidase inhibitor, antibacterial (*B. subtilis*) antibacterial (*E. faecalis, S.aureus,* MRSA) antibacterial (*S. aureus*)	[20] [54] [74]
Bacillisporin C	antiproliferative (K562, HL-60, BGC-823, HeLa)	[104]
Bacillisporin F	protein tyrosine phosphatase inhibitor	[87]
Bacillisporins K–L	antibacterial (*S. aureus*)	[74]
Bromothiobenzoic acid	antibacterial (*E. coli, K. pneumoniae, S. aureus*), cytotoxic (HCT 116, HepG2 MCF-7)	[29]
Chrodrimanins A, C	antioxidant	[84]
Chrodrimanin B	protein tyrosine phosphatase inhibitor antioxidant	[101] [84]
Chrodrimanins K, N	antiviral (H1N1)	[21]
Chrodrimanins O, R–S	protein tyrosine phosphatase inhibitor	[101]
Chromosulfine	antiproliferative-proapoptotic (MCF-7, K562, HL-60, HeLa, BGC-823)	[102]
Citrorosein, questinol	lipid lowering	[83]
Conioxanthone A	immunosuppressive	[17]
Corymbiferan lactone A	antiproliferative (HL-60, BGC-823, HeLa)	[104]
Cyclosecosteroid A	acetylcholineterase inhibitor	[75]
Cyclotryprostatin B	anti-inflammatory	[47]
Dankasterone	antiproliferative (HL-60, A549, MCF-7, SW480)	[109]
Dehydroisopenicillide	anticholesterol, lipid lowering	[19]
Dihydro-hydroxy-hydroxymethyl-methoxy-methyl-naphthofurandione	anti-inflammatory	[68]
Dihydroxyconiothyrinone B	antibacterial (*E. coli, E. tarda, S. aureus*), antioxidant	[93]
Dihydroxy-dimethyl-dibenzofuran	antibacterial (*E. coli, E. faecalis,* MRSA, *S. aureus*), antifouling (*B. neritina*)	[89]
Dihydroxyfumitremorgin C	anti-inflammatory	[47]
Dihydroxy-hydroxybenzylidene-methylbutenyl-indane-carboxylic acid methyl ester	antibacterial (*B. cereus*, *S. albus, S. aureus*)	[28]
Dihydroxy-hydroxypentyl-isochromanone	α-glucosidase inhibitor α-glucosidase inhibitor, antioxidant	[32] [40]
Dihydroxy-hydroxypropyl-methyl-isochromenone, hydroxy-hydroxypropyl-methoxyisochromanone, hydroxypropyl-hydroxy-dihydroisocoumarin	antioxidant	[40]
Dihydroxyisocupressic acid	antibacterial (*V. parahemolyticus*)	[72]
Dihydroxy-methoxy-methylisochromenone, dihydroxy-pentyl-hydroxy-isochromanone, dimethyl-dihydroxyisocoumarin, hydroxy-hydroxyethyl-methoxyisocoumarin, hydroxy-hydroxymethyl-methoxy-methylisocoumarin, hydroxy-methoxy-dimethylisocoumarin	α-glucosidase inhibitors	[32]
Dihydroxy-methyldendryol E	antibacterial (*S. aureus*), antioxidant	[93]
Dihydroxy-methyl-hydroxymethyl-xanthone	antibacterial (*A. hydrophila*)	[45]
Diolhinokiic acid	antiproliferative (HL-60, A549)	[109]
Diorcinol, methoxycarbonyldiorcinol	antibacterial (*E. coli, E. faecalis,* MRSA, *S. aureus*), antifouling (*B. neritina*)	[89]
Ditalaromylectone A	antifungal (*C. albicans*)	[51]
Epoxyberkeleydione	lipid lowering	[59]
Ergosta-trienol	cytotoxic (HepG2, MCF-7)	[29]
Ethyl-dimethoxyjuglone	anti-inflammatory	[68]
Eurothiocin A	α-glucosidase inhibitor anti-inflammatory	[76] [38]
Eurothiocins D, F, G	α-glucosidase inhibitors	[48]
Expansols E–F	antifouling (*B. neritina*)	[89]
Fructigenines A–B	antiproliferative (K562, HeLa, HL-60, BGC-823, MCF-7)	[107]
Fumigaclavine C, fumigatin oxide, fumiquinazoline F, fumiquinone B	anti-inflammatory	[47]
Funicone, deoxyfunicone, 3-*O*-methylfunicone, hydroxyvermistatin, methoxyvermistatin	anticholesterol, lipid lowering	[19]
Fusarielins M, O, P	cytotoxic (B16)	[73]
Hydroxyconiothyrinone B	antibacterial (*S. aureus*), antioxidant	[93]
Hydroxy-ergosta-trienone	antioxidant	[84]
Hydroxy-hydroxyethyl-dimethoxy-naphthalenedione	anti-inflammatory, cytotoxic (RAW 264.7)	[68]
Hydroxy-hydroxy-hydroxymethyl-methylphenoxy-benzoic acid	protein tyrosine phosphatase inhibitor	[22]
Hydroxy-methoxy-benzofuranyl-ethanone, hydroxy-methoxy- methylbenzofuran-carboxylic acid	antibacterial (*B. subtilis, E. coli, S. aureus, S. epidermidis*)	[32]
Hydroxy-methyl-dimethoxycoumarin	antibacterial (MRSA), antifungal (*C. albicans*)	[70]
Hydroxypentacecilide A	antiviral (H1N1)	[21]
Hydroxypropyl-methyl-dihydroxychromone	antibacterial (*Salmonella*)	[20]
Isobutyric acid dihydroxy-methyl-oxo-dihydro-naphththalenyl methyl ester	α-glucosidase inhibitor	[32]
Isocyclopaldic acid	antibacterial (*A. hydrophila, E. coli, M. luteus, P. aeruginosa, V. anguillarum, V. harveyi, V. parahemolyticus*)	[45]
Isotryptoquivaline F	anti-inflammatory	[47]
Javanicin	anti-inflammatory, cytotoxic (RAW 264.7)	[68]
Macrosporusone D	antibacterial (*S. aureus*)	[74]
*epi*-Mangrovamide A	antibacterial (*V. harveyi*. *V. parahaemolyticus*)	[45]
Mangrovamide I	antibacterial (*A. hydrophila, E. coli, M. luteus, P. aeruginosa, V. anguillarum, V. harveyi, V. parahemolyticus*)	[45]
Methoxy-methyl-biphenyl-triol	antibacterial (*S. aureus*)	[51]
Methylhexahydrofuro-furanylethanol	cytotoxic (SF-268, MCF-7, HepG2, A549)	[96]
Methylincisterol, dimethylincisterol A3	antifouling (*B. neritina*)	[89]
Methylpseurotin A, norpseurotin A	anti-inflammatory	[47]
Methyltenellate	lipid lowering	[19]
Monodictyphenone	protein tyrosine phosphatase inhibitor	[22]
Nafuredin B	cytotoxic (HeLa, MCF-7, K562, HCT 116, HL-60, A549)	[24]
Penicidone C	α-glucosidase inhibitor	[26]
Penicifuran A	antibacterial (*E. coli*, MRSA), antifungal (*C. albicans*)	[70]
Peniciisocoumarins C, F, G	antioxidant	[40]
Peniciisocoumarin D	α-glucosidase inhibitor, antioxidant antibacterial (*E. coli*, MRSA), antifungal (*C. albicans*)	[40] [70]
Peniciisocoumarin E	antioxidant	[30]
Peniciisocoumarin H	antibacterial (*E. coli*, MRSA), antifungal (*C. albicans*)	[27]
Penicillide	α-glucosidase inhibitor cytotoxic (H1975, HL7702, K562, MCF-7)	[26] [23]
Penicilloxalone B	antibacterial (*E. coli*, MRSA) antioxidant	[70] [30]
Penicimarins B–C	α-glucosidase inhibitor	[32]
Penicimarin G	antibacterial (*B. cereus, E. coli, S. aureus*), antioxidant α-glucosidase inhibitor, antioxidant	[92] [30]
Penicimarin H	antioxidant α-glucosidase inhibitor, antioxidant	[40] [30]
Penicimarin I	α-glucosidase inhibitor	[30]
Penicimarins L–M	antioxidant	[30]
Penicimarin N	α-glucosidase inhibitor, antioxidant	[40]
Penicimumide	antiproliferative (A549, HeLa, MCF-7, HepG2, NCI-H1975, HL-60, K562, LS180, SW480, HT29, BXPC-3, PANC-1)	[56]
Penicimutalidine	antiproliferative (K562, HL-60, BGC-823, HeLa)	[104]
Penicimutamides A–F	antiproliferative (K562, HL-60, BGC-823, HeLa)	[105,106]
Penicimutanines A, C	antiproliferative (K562, HeLa, HL-60, BGC-823, MCF-7)	[107]
Penicimutanolones A–B, penicimutanolone A methyl ether	antiproliferative (A549, HeLa, MCF-7, HCT 116, HepG2, NCI-H1975, HL-60, K562, LS180, SW480, HT29, PC-3, BXPC-3, PANC-1)	[56]
Penicimutide	antiproliferative (HeLa)	[18]
Penioxalicin	antibacterial (MRSA)	[80]
Peniphenone, pinselin	immunosuppressive	[17]
Penixanthones A–B	antiallergic cytotoxic (H1975, HL7702, K562, MCF-7)	[25] [23]
Penixanthones C–D	cytotoxic (K562, MCF-7, Huh7)	[90]
Pensulfonamide	antibacterial (*E. coli, K. pneumoniae, S. aureus*), antifungal (*A. niger* and *C. albicans*), cytotoxic (MCF-7, HCT 116, HepG2)	[29]
Pensulfonoxy	antibacterial (*E. coli, K. pneumoniae, S. aureus*), antifungal (*A. niger*), cytotoxic (HCT 116, HepG2)	[29]
Pestalotiorin	α-glucosidase inhibitor	[40]
Propenyl-octene-triol	cytotoxic (SF-268, MCF-7, HepG2, A549)	[96]
Purpactin A	antibacterial (*H. pylori*) α-glucosidase inhibitor	[77] [82]
Purpurides E–F	antibacterial (*E. coli*, MRSA), antifungal (*C. albicans*)	[27]
Purpuride G	antibacterial (*E. coli*, MRSA), antifungal (*C. albicans*), antiproliferative (U251, U87MG)	[27]
Purpurogenic acid	antiproliferative (K562, HL-60, HeLa, BGC-823)	[99]
Roussoellol C	antiproliferative (HL-60, A549, MCF-7, SW480)	[109]
Rugulosin A	antibacterial (*S. aureus*)	[74]
Rugulosuvine	antiproliferative (K562, HeLa, HL-60, BGC-823, MCF-7)	[107]
Sch1385568	antibacterial (MRSA, *S. aureus*)	[54]
Sch725680	α-glucosidase inhibitor, antibacterial (*M. smegmatis*, *S. aureus*)	[26]
Secalonic acid D	antibacterial (*H. pylori*), cytotoxic (Bel-7402, HCT 116)	[77]
Secopenicillide A	lipid lowering	[19]
Secopenicillide B	antibacterial (*H. pylori*)	[77]
Sequoiamonascin C	cytotoxic (B16, MCF-7)	[73]
Sescandelin	α-glucosidase inhibitor antibacterial (*E. coli*, MRSA), antifungal (*C. albicans*)	[32] [70]
Sescandelin B	α-glucosidase inhibitor	[32]
SF226	antiproliferative (K562, HL-60, BGC-823, HeLa)	[104]
Solaniol	anti-inflammatory, cytotoxic (RAW 264.7)	[68]
Sydowinin A	immunosuppressive	[17]
Talabenzofuran C	α-glucosidase inhibitor	[76]
Talacyanol A	anti-inflammatory, cytotoxic (HCT-15, NUGC-3, MDA-MB-231, PC-3, NCI-H23, ACHN)	[38]
Taladrimanin A	antibacterial (*S. aureus*), antiproliferative-proapoptotic (MGC803, MKN28)	[78]
Talamin A	antibacterial (*V. vulnificus*), antioxidant	[52]
Talamin B	antibacterial (MRSA, *V. vulnificus*)	[52]
Talamin D	antioxidant	[52]
Talanaphthoquinone A	anti-inflammatory, cytotoxic (RAW 264.7)	[68]
Talaramide	mycobacterial PknG kinase inhibitor	[91]
Talarocyclopenta A	antibacterial (*E. coli*, *S. aureus*), anti-inflammatory	[34]
Talarocyclopenta B	antibacterial (*B. cereus, B. subtilis, E. coli, M. tetragenus, S. albus, S. aureus*), anti-inflammatory	[34]
Talarocyclopenta C	anti-inflammatory	[34]
Talarodilactones A–B	cytotoxic (L5178Y)	[63]
Talarodrides A–B	antibacterial (*P. mirabilis*, *V. parahemolyticus*)	[72]
Talaroisocoumarin A	antibacterial (*E. coli*, MRSA), antifungal (*C. albicans*)	[70]
Talaromarin F	antioxidant	[40]
Talarominine A	antibacterial (MRSA, *M. luteus, P. aeruginosa, V. harveyi*, *V. vulnificus*), antioxidant	[52]
Talaromynoid E	protein tyrosine phosphatase inhibitor	[59]
Talaromynoids G–I	lipid lowering	[59]
Talaromyolides D, I, K	antiviral (PRV)	[58,103]
Talaromyone A	antibacterial (*H. pylori*)	[77]
Talaromyone B	antibacterial (*B. subtilis*), α-glucosidase inhibitor	[82]
Talaromyoxaones A–B	protein tyrosine phosphatase inhibitors	[60]
Talaronin E	antibacterial (*H. pylori*)	[77]
Talaropeptins A–B	antifungal (*F. oxysporum*)	[108]
Talascortenes	antibacterial (*A. hydrophila, E. coli, E. tarda, M. luteus, P. aeruginosa, V. harveyi, V. parahemolyticus*), antifungal (*C. gloeosporioides, F. oxysporum, G. graminis, R. cerealis*)	[64]
Talasteroid	antioxidant	[84]
Talaverrucin A	Wnt/β-catenin pathway inhibitor	[81]
Tenellic acid A	α-glucosidase inhibitor	[82]
Tetrahydroxymethylbenzophenone, trihydroxymethylxanthone	antibacterial (*E. coli, P. aeruginosa, S. aureus, V. alginolyticus, V. harveyi, V. parahaemolyticus*), antioxidant	[49]
Trihydroxybutyl-hydroxy-hydroxy-methylphenoxy-methylphenylacetate	antibacterial (*E. coli, E. faecalis,* MRSA, *S. aureus*)	[89]
Trihydroxy-hydroxyethyl-isocoumarin	α-glucosidase inhibitor	[32]
Trihydroxy-methoxymethylbenzophenone	antioxidant	[49]
Vermistatin	anticholesterol, lipid lowering cytotoxic (B16)	[19] [73]
Verruculide B_2_	antibacterial (*S. aureus*)	[21]
Verruculosin A, xenoclauxin	protein tyrosine phosphatase inhibitor	[87]

^1^ Microbial species and cell types used in bioassays are indicated in brackets.

## Data Availability

No new data were created.

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
