# Peer review of "The Outstanding Chemodiversity of Marine-Derived Talaromyces"

_biomolecules, 2023, doi:10.3390/biom13071021_

Round 1

Reviewer 1 Report

This review, dedicated to the Talaromyces species and more particularly to the metabolites produced by marine derived strains, constitutes a consistently well-structured and well-written work. This summary of articles published since 2016 is a rigorous and interesting account of recent publications. Thank you to authors for this nice work.

Author Response

Thank you very much for your positive comments on our work.

Reviewer 2 Report

This is a comprehensive review of the literature on Telaromyces natural products published since 2016. The groupings are clearly described in appropriate Tables. All discussion is relevant and, where appropriate, discusses the shortcomings and potential issues with the comparison of the work of many diverse groups of researchers.

Overall the manuscript is very well written. There are a few minor issues with the language which I highlight here:

Table 1: Reference 21 - "annelide" should be "annelid"

Line 75: "a various assortment of plant and animal organisms" can be simplified to "a variety of plants and animals"

Line 89: change "tentative" to "attempt"

Lines 94: suggested revision "The remarkable number of new compounds resulting from the biochemical characterization of Talaromyces strains shows some degree of specificity, possibly reflecting chemotaxonomic relevance"

Lines 98-100: suggested revision "however, after many years of study some compounds have been found exclusively or almost exclusively in Talaromyces and can be considered as candidates in the assessment of phylogenetic relations"

Line 134: "result to be able to produce" change to "may be able to produce"

Lines 153-154: "Structural elucidation has also ... in other classes."

Lines 215-216: The finding of a series... differing in certain... has allowed comparative hypotheses concerning bioactivities"

Line 217: "in which hydroxyl"

Line 243: Change "On different instances" to "In another case"

Line 348-9: A suggestion - I think this is what is implied "... in both qualitative and quantitative terms, however the preliminary nature of many of the studies does not allow a determination of the applicability of the discoveries in terms of future development. "

Line 350: "respondent" should be "responsive"?

Lines 389-391: "... published in the last seven years has resulted in the impressive number ... extracted from cultures of Talaromyces strains."

Line 399: "The extent to which..."

Author Response

Thank you very much for your positive comments on our work, and for the accurate revision of the text. All the suggested adjustments have been made in the updated version of the manuscript.